# Lymphadenopathy after the Anti-COVID-19 Vaccine: Multiparametric Ultrasound Findings

**DOI:** 10.3390/biology10070652

**Published:** 2021-07-12

**Authors:** Giulio Cocco, Andrea Delli Pizzi, Stefano Fabiani, Nino Cocco, Andrea Boccatonda, Alessio Frisone, Antonio Scarano, Cosima Schiavone

**Affiliations:** 1Unit of Ultrasound in Internal Medicine, Department of Medicine and Science of Aging, “G. d’Annunzio” University, 66100 Chieti, Italy; sfabiani92@gmail.com (S.F.); cschiavone@unich.it (C.S.); 2Department of Neurosciences, Imaging and Clinical Sciences, “G. d’Annunzio” University, 66100 Chieti, Italy; andreadellipizzi@gmail.com; 3Departmental Faculty of Medicine and Surgery, Campus Bio-Medico University, 00128 Rome, Italy; ninococco@gmail.com; 4Department of Internal Medicine, University of Bologna, 40010 Bologna, Italy; andrea.boccatonda@gmail.com; 5Department of Innovative Technologies in Medicine & Dentistry, “G. d’Annunzio” University, 66100 Chieti, Italy; ales.frisone@gmail.com (A.F.); ascarano@unich.it (A.S.)

**Keywords:** lymphadenopathy, anti-COVID-19 vaccines, ultrasound

## Abstract

**Simple Summary:**

Post-anti-COVID-19 vaccine lymphadenopathy is a not uncommon event. In this study, we investigated the multiparametric ultrasound findings of patients with post-vaccine lymphadenopathy and compared these findings among different anti-COVID-19 vaccines. We evaluated patients presenting with post-anti-COVID-19 lymphadenopathy. The presence, size, location, number, morphology, cortex–hilum, superb microvascular imaging and elastosonography of lymph nodes were assessed. They were axillary and supraclavicular ipsilateral to the injection site. Prevalent ultrasound features included oval morphology, asymmetric cortex with hilum evidence, central and peripheral vascular signals at superb microvascular imaging and elastosonography patterns similar to the surrounding tissue. We found no significant differences between the three COVID-19 vaccines: the Pfizer/BioNTech BNT162b2 mRNA vaccine, the AstraZeneca ChAdOx1 vaccine and Moderna’s mRNA-1273 vaccine. Some ultrasound lymph node features, such as round morphology, no hilum evidence and hard pattern, may mimic pathological lymph nodes. An awareness of the patient’s history (vaccine injection and oncological history) and ultrasound findings may help in the early recognition of this clinical scenario and in the appropriate selection of patients for a short-term US follow-up.

**Abstract:**

*Background*: Post-anti-COVID-19 vaccine lymphadenopathy has recently been described in the literature. In this study, we investigated the multiparametric US findings of patients with post-vaccine lymphadenopathy and compared these findings among different anti-COVID-19 vaccines. *Methods*: We retrospectively evaluated 24 patients who underwent US between January and May 2021 due to post-anti-COVID-19 lymphadenopathy. The presence, size, location, number, morphology, cortex-hilum, superb microvascular imaging (SMI) and elastosonography of lymph nodes were assessed. Descriptive statistics were calculated and differences among anti-COVID-19 vaccines were analyzed using the Kruskal–Wallis test. A *p*-value ≤ 0.05 was considered statistically significant. *Results*: Sixty-six nodes were assessed. They were axillary (mean 1.6 cm ± 0.16) in 11 patients (45.8%) and supraclavicular (mean 0.9 cm ± 0.19) in 13 patients (54.2%). In 20 patients (83.3%), the number of nodes was ≤3. Prevalent US features included oval morphology (18, 75%), asymmetric cortex with hilum evidence (9, 37.5%), central and peripheral vascular signals (12, 50%) at SMI and elastosonography patterns similar to the surrounding tissue (15, 71.4%). No significant differences among the three anti-COVID-19 vaccines were observed (*p* > 0.05). *Conclusions*: Anti-COVID-19 vaccines may present lymphadenopathy with “worrisome” US features regarding size, shape, morphology, cortex-hilum, SMI and elastosonography. An awareness of the patient’s history and US findings may help in the early recognition of this clinical scenario and in the appropriate selection of patients for a short-term US follow-up.

## 1. Introduction

Since March 2019, the world has been shocked by the pandemic caused by COVID-19, which has caused millions of deaths and widespread economic and social damage [1]. COVID-19 was first detected in December 2019 in Wuhan, China and causes a highly infectious disease [2,3].

Once infected, patients usually show an extremely variable clinical course, ranging from mild symptoms (fever and cough) to bilateral interstitial pneumonia. In the most severe cases, infection progresses into acute respiratory distress syndrome (ARDS) with diffuse alveolar consolidations (diffuse patchy-like lesions) [4].

Despite periodic lockdowns and improved therapeutic strategies, vaccines currently represent the most efficient means to control and stop the COVID-19 pandemic [1]. Although vaccines are fairly safe drugs, they are not completely risk-free and adverse events may occur following vaccination [5].

SARS-CoV-2, while reproducing, develops mutations, resulting in variants different from the original strain. The B.1.1.7 variant was first described in the United Kingdom in late December 2020 and, subsequently, the B.1.351 variant was reported in Africa [6]. A third variant, B.1.1.248/B1.1.28/P1, was reported in Brazil in early January 2021, and more recently, the B.1.427/B.1.429 lineage was identified in California [6]. The BNT162b2 vaccine demonstrated effectiveness against variant infection in Qatar, with an effectiveness of 89.5% and 75% at 14 or more days after the second dose [7].

Following the recent approval of the US Food and Drug Administration (FDA) and the rollout of anti-COVID-19 vaccines, there were several cases of lymphadenopathy. Moreover, recent articles have reported cases of unilateral axillary and supraclavicular adenopathy [8,9,10,11,12,13,14].

Sometimes, post-COVID-19 vaccine supraclavicular and axillary lymph nodes may mimic pathological lymph nodes. Ultrasound (US) examination may represent the first-line imaging method due to its speed, low cost and repeatability and has already played a central role in this pandemic [15,16,17]. Sonologists should be aware that a recent anti-COVID-19 vaccine can present an etiology of supraclavicular and axillary lymph nodes with suspicious US features. In this retrospective study, we investigated the multiparametric US findings of patients with post-vaccine lymphadenopathy and compared these features among three different anti-COVID-19 vaccines.

## 2. Materials and Methods

### 2.1. Study Population

This is a retrospective, observational, spontaneous and autonomous study for which the authorization of the ethics committee was waived. Patient data were obtained in accordance with National Privacy Regulations (https://www.privacy-regulation.eu/en/, accessed on 1 June 2021). We retrospectively included a total of 28 consecutive patients who underwent a clinically indicated post-anti-COVID-19 US exam between January 2021 and May 2021 due to lymphadenopathy. Patients were included if they met all the following criteria: (a) they received at least one dose administration of an anti-COVID-19 vaccine, (b) they underwent a clinically indicated US due to post-vaccine lymphadenopathy observed at clinical examination (palpable node, pain and swelling) or due to inconclusive radiological examination, including computed tomography (CT), magnetic resonance (MR) and X-ray, showing pathologically enlarged nodes. Four patients were excluded because they had fever and refused the US examination. The final study population was composed of 24 patients. No patients with oncohematologic or autoimmune disease were included in our study.

### 2.2. US Protocol

Ultrasound exams were performed by 3 dedicated sonologists, using a Canon Aplio I800 (Japan) in combination with a linear 5 to 18 MHz array matrix probe. All the sonologists recorded the data in consensus. In detail, they assessed for each patient the presence, size, location, number, morphology (round or oval) and cortex–hilum (simmetric cortex with hilum evidence, asimmetric cortex with hilum evidence, no hilum evidence) of the lymph nodes [18,19]. Moreover, superb microvascular imaging (SMI) and elastosonography were evaluated [20]. In detail, SMI investigated the presence of centrally located vascular hilum without aberrant vascular signals, peripheral vascular signals or central and peripheral vascular signals. The elasticity assessment of the lymph nodes was done by evaluating two regions of interest (ROIs): one ROI positioned on the target region (lymph node) and the second ROI on the adjacent tissue (normal muscles or subcutaneous tissue). The strain ratio and the subsequent differentiation into “soft” and “hard” was then automatically computed by the USG device [21,22]. A dedicated flowchart summarizing the methodology is shown in Figure 1.

### 2.3. Statistical Analysis

Distribution normality was tested using the Shapiro–Wilk test. Descriptive statistics were calculated for all patients. Differences in US findings according to different anti-COVID-19 vaccines were analyzed using the Kruskal–Wallis test. All the statistical analyses were performed using IBM SPSS Statistic software, version 20 (IBM, Armonk, NY, USA). A *p* value ≤ 0.05 was considered statistically significant.

## 3. Results

We examined 24 patients and a total of 66 lymph nodes (Table 1). All the lymph nodes were ipsilateral to the vaccine injection site. Three anti-COVID-19 vaccines were administered: Pfizer/BioNTech BNT162b2 mRNA vaccine (13, 54.2%), AstraZeneca ChAdOx1 vaccine (8, 33.3%) and Moderna’s mRNA-1273 vaccine (3, 12.5%). Seventeen (70.8%) patients developed lymphadenopathy after the first dose and seven patients (29.2%) after the second dose. The prevalent localization was axillary in 11 patients (45.8%) and supraclavicular in 13 patients (54.2%). The supraclavicular lymph nodes never exceeded 1.5 cm in size and the median size was 1.6 cm ± 0.16, while axillary lymph nodes never exceeded 3 cm in size and the median size was 0.9 cm ± 0.19. In 20/24 patients (83.3%), the number of nodes was ≤3. The prevalent morphology was ovular in 18 patients (75%) and round in six patients (25%). The prevalent cortex–hilum pattern was asymmetric cortex with hilum evidence (9, 37.5%), followed by the absence of hilum (8, 33.3%) and symmetric cortex with hilum evidence (7, 29.2%). The SMI showed a prevalence of central and peripheral vascular signals (12, 50%) and centrally located vascular hilum without aberrant vascular signals (11, 45.8%); only one case (4.2%) with peripheral vascular signals was observed. Elastosonography patterns were similar to the surrounding tissue in 15 cases (71.4%) and prevalently hard in nine cases (28.6%). There were no significant differences in the US features among the three anti-COVID-19 vaccines (*p* > 0.05). All patients underwent a short-term follow-up ultrasound (2 weeks). We repeated the ultrasound examination every 2 weeks, until the lymph nodes normalized their features. When the lymph nodes returned to normality, we stopped the follow-up. Seven (29.2%) patients showed US normalization of lymph node characteristics within 30 days, three (12.5%) patients within 45 days and two (8.3%) patients within 60 days; 12 (50%) patients did not require US follow-up. The average number of days needed for normalization was 26.9 ± 14.7. None of the patients required biopsy, fine needle aspiration or other second-level diagnostic tests after the US follow-up (Figure 2, Figure 3, Figure 4, Figure 5, Figure 6, Figure 7 and Figure 8).

## 4. Discussion

In our study, we found that the Pfizer/BioNTech BNT162b2 mRNA, AstraZeneca ChAdOx1 and Moderna mRNA-1273 vaccines can induce axillary and supraclavicular lymphadenopathy, ipsilateral to the injection site. Our multiparametric US assessment revealed no significant differences in the US features among the three anti-COVID-19 vaccines. In our study, lymphadenopathy was observed after both the first and the second vaccine doses [9]. Not infrequently, the appearance of lymphadenopathies post-COVID-19 vaccines represented a diagnostic challenge and dilemma, with abnormal lymph nodes [23]. In fact, in some of the examined patients, we observed “worrying” features, usually suspicious for malignancy, such as round morphology (6, 25%) hilum absence (8, 33.3%) and asymmetrical cortex (9, 37.5%).

The majority of patients underwent US examinations for pain and/or palpable masses in supraclavicular or axillary sites, or less frequently for asymptomatic lymph nodes found incidentally with other imaging tests (mammography, CT or MRI). We found lymphadenopathy reaction after the anti-COVID-19 vaccine in every age group (from 25 to 74 years old). We observed a lymphadenopathy reaction after the anti-COVID-19 vaccine, especially after the first dose and less frequently in the second dose. The US lymph node feature normalization was variable. We followed-up with patients for 60 days. We had four patients with an oncological history: two with breast cancer, one with kidney cancer and one with melanoma. Our study did not include patients with onco-hematological/autoimmune diseases, and the US pattern in these patients may be different. Further larger studies, possibly including patients with autoimmune/onco-hematological diseases, are necessary to assess if they present with more specific US findings and how the US scenario changes during the follow-up. This approach may be crucial for the development of a dedicated clinical algorithm.

Anti-COVID-19 vaccines are not the first vaccines to document a lymphadenopathy reaction, but it has been documented in the literature as occurring shortly after receiving the smallpox, Bacille Calmette–Guerin (BCG), human papillomavirus (HPV) and H1N1 influenza A virus vaccines [24,25,26]. Generally, hyperplastic axillary and supraclavicular lymph nodes are more common after a vaccine that evokes a strong immune system response [24,25,26]. Morphology, hilum features and cortical thickness of the lymph nodes are the most important criteria for distinguishing between normal and abnormal lymph nodes. Cortical thickness >3 mm, round morphology and encroachment on or displacement of the hyperechoic hilum are often suggestive of a pathologic process [23]. A lymphadenopathy reaction can be found incidentally on imaging tests, such as routine screening or cancer surveillance (mammography, CT or MRI scans). However, US is the preferred imaging modality for evaluating axillary lymph nodes [27,28]. For these reasons, a US examination will be often required. In some patients, the lymph nodes at US had a round morphology, with no hilum evidence and a hard pattern on elastosonography. These US features, especially in patients under cancer surveillance, could be alarming [23,29].

The Society of Breast Imaging (SBI), to prevent the diagnostic dilemma of vaccine-induced lymphadenopathy, advises to consider scheduling screening exams prior to the first dose of a COVID-19 vaccination or 4–6 weeks following the second dose of a COVID-19 vaccination [30]. Oncology patients are generally advised to be vaccinated against COVID-19, particularly because they are at higher risk of dying from COVID-19 than the general population [31,32,33]. To avoid confusion for patients undergoing treatment for cancer in one breast, the COVID-19 vaccine shot should be given in the arm on the other side. The vaccine can also be injected into the thigh to prevent any issues with lymph node swelling [29,30].

A case series by Mehta et al. considered four patients that received the Pfizer and Moderna vaccines. In case 1, the patient reported a self-detecting ipsilateral and unilateral axillary adenopathy 9 days after receiving the first dose of the Pfizer-BioNTech COVID-19 vaccine. Unilateral axillary adenopathy was incidentally noted in case 2 and case 4; therefore, the exact onset of this reaction after receiving the COVID-19 vaccine remains unclear in these cases. The time between receiving the Pfizer-BioNTech COVID-19 vaccination and detection of unilateral axillary adenopathy was in keeping with the average duration of adenopathy reported by the Centers for Disease Control and Prevention (CDC). However, the time between receiving the Moderna COVID-19 vaccination and detection of unilateral axillary adenopathy in case 3 was 13 days, much longer than the average duration of 1–2 days reported by the CDC in recipients of the Moderna COVID-19 vaccine [9].

Furthermore, a case series by Özütemiz et al. showed the US lymph node features of five patients that received the Pfizer vaccine. In two cases, there was a pathologic confirmation of benign reactive lymphadenopathy secondary to vaccination, although the remaining three cases were attributed to recent vaccine administration without confirmation with histopathological evaluation [10].

A paper by Granata et al., describing a population of 18 patients that received the Pfizer vaccine, found that 43.1% of lymph nodes showed eccentric cortical thickening with a wide echogenic hilum and oval shape, and 32.8% of lymph nodes showed asymmetric eccentric cortical thickening with a wide echogenic hilum and oval shape. A total of 17.2% of lymph nodes showed concentric cortical thickening with a reduction in the width of the echogenic hilum and oval shape, and 6.9% showed a huge reduction in and displacement of the echogenic hilum and a round or oval shape [14]. These results from the current literature demonstrate the heterogeneity of US features that can be found after COVID-19 vaccines and that there can be patterns that mimic malignant lymph nodes. Our study has some limitations. First of all, it is a retrospective study involving a relatively small and spontaneous sample; consequently, we did not know the real incidence of lymphadenopathy. In this regard, the real incidence of lymphadenopathy was not calculable. Further prospective and possibly multicenter studies are needed to confirm our findings. Finally, the study was limited by the absence of a pathological correlation.

## 5. Conclusions

All three anti-COVID-19 vaccines may present lymphadenopathy with “worrisome” US features regarding the size, shape, morphology, cortex–hilum, SMI and elastography of lymph nodes. An awareness of the patient’s history and US findings may help sonologists to recognize this clinical scenario early and appropriately select patients for a short-term US follow-up.

## Figures and Tables

**Figure 1 biology-10-00652-f001:**
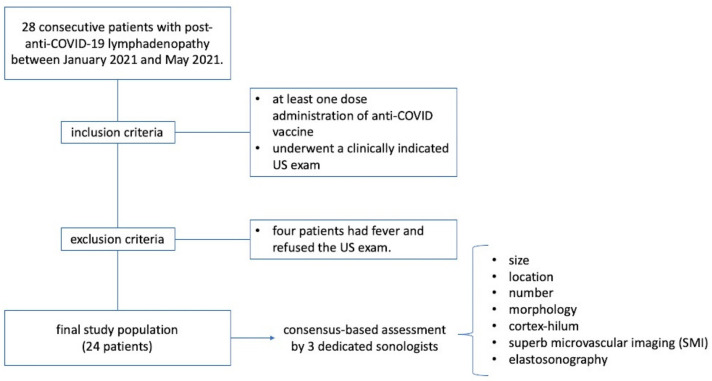
Flowchart summarizing methodology.

**Figure 2 biology-10-00652-f002:**
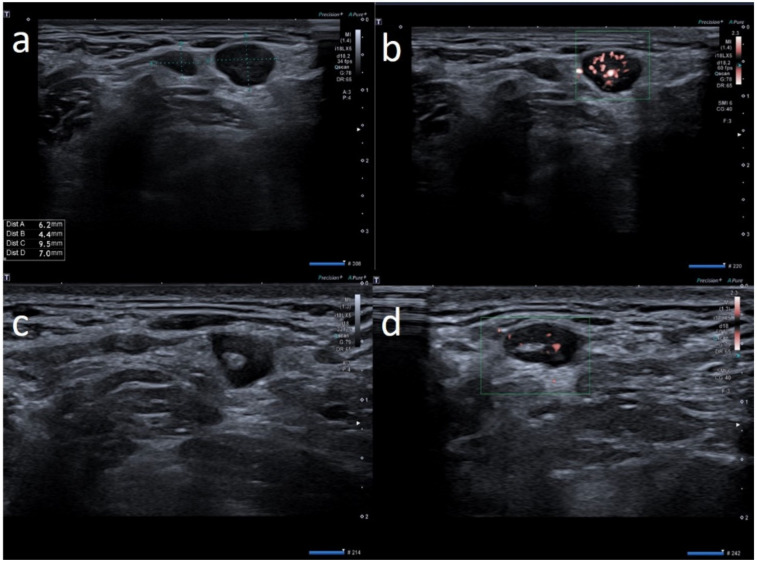
28-year-old female with palpable unilateral sopraclavicolar adenopathy noted 3 days after receiving the first dose of the AstraZeneca COVID-19 vaccine in her left deltoid muscle. (**a**) B-mode sonogram image shows 2 round hypoechoic lymph nodes with hilum absence. (**b**) SMI images show central and peripheral vascularization. (**c**) After 2 weeks, sonogram image shows the restoration of normal pattern of the smaller lymph node whereas the other node presents asymmetric cortical thickening but with hilum reappearance. (**d**) After 1 month, the cortical thickening appears uniform, with hilum evidence and normal vascularization.

**Figure 3 biology-10-00652-f003:**
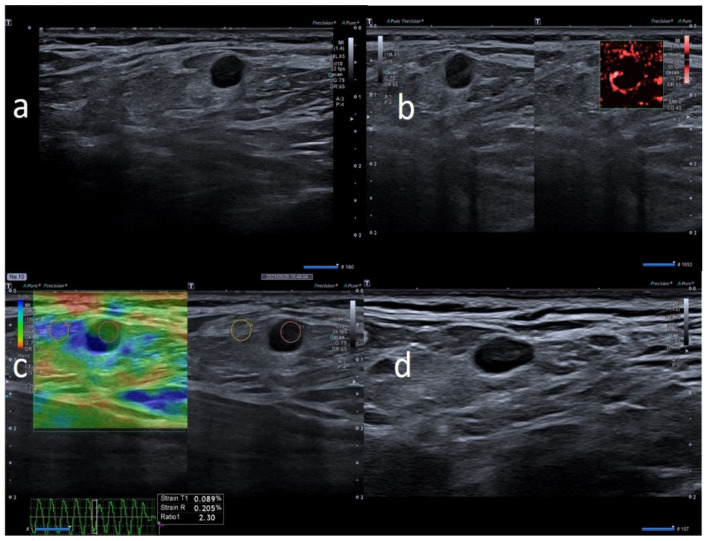
74-year-old female with palpable unilateral sopraclavicolar adenopathy noted 2 days after receiving the first dose of the AstraZeneca COVID-19 vaccine in her left deltoid muscle. (**a**) B-mode sonogram image shows round, hypoechoic lymph node with hilum absence. (**b**) SMI images shows peripheral vascularization. (**c**) Elastosonography strain shows the hard pattern of the node surrounding tissue. (**d**) After 1 month, the cortical thickening appears uniform, with hilum evidence and normal vascularization.

**Figure 4 biology-10-00652-f004:**
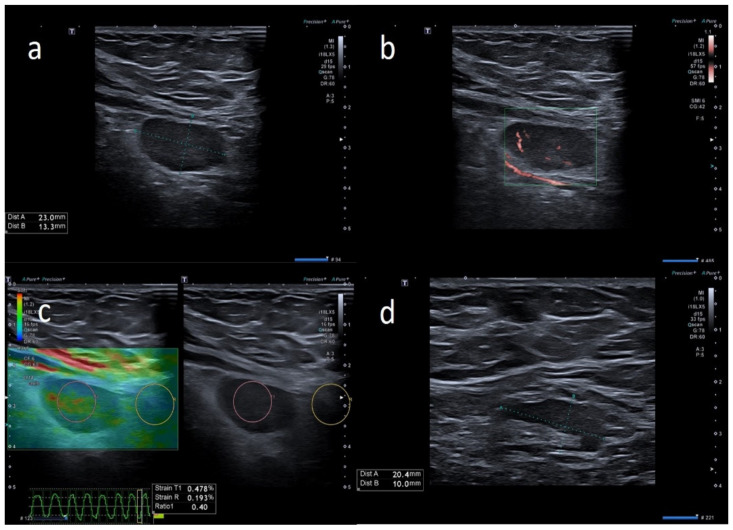
49-year-old female with unilateral left axillary adenopathy noted 6 days after receiving the first dose of the Pfizer-BioNTech COVID-19 vaccine in her left deltoid muscle. (**a**) B-mode sonogram image shows ovular, hypoechoic lymph node with hilum absence. (**b**) SMI images show central and peripheral vascularization. (**c**) Elastosonography strain shows similar pattern of the node compared to surrounding tissue. (**d**) After 1 month, the cortical thickening appears uniform, with hilum evidence.

**Figure 5 biology-10-00652-f005:**
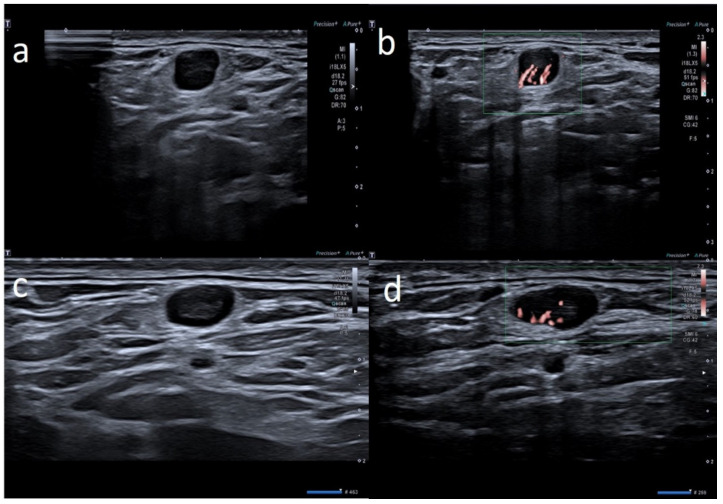
35-year-old male with palpable unilateral sopraclavicolar adenopathy noted 3 days after receiving the first dose of the Pfizer-BioNTech COVID-19 vaccine in his left deltoid muscle. (**a**) B-mode sonogram image shows a round, hypoechoic lymph node with hilum absence. (**b**) SMI images show central and peripheral vascularization. (**c**) After 2 weeks, sonogram image shows ovular morphology, asymmetric cortical thickening but with faint hilum reappearance. (**d**) After 1 month, the cortical thickening appears uniform, with hilum evidence and normal vascularization.

**Figure 6 biology-10-00652-f006:**
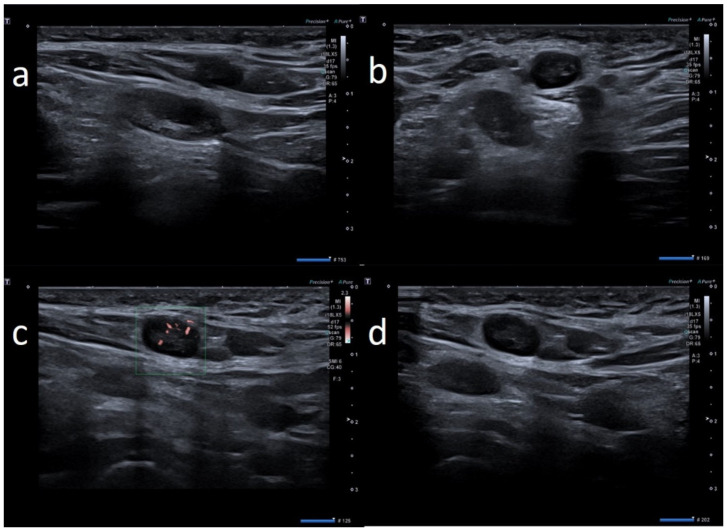
25-year-old female with unilateral left axillary adenopathy noted 5 days after receiving the first dose of the Pfizer-BioNTech COVID-19 vaccine in her left deltoid muscle. (**a**,**b**) B-mode sonogram image shows ovular lymph nodes with asymmetric cortex and dislocate hilum. (**c**) After 2 months, SMI image shows normal vascularization (**d**) and cortical thickening appears uniform with normal hilum localization.

**Figure 7 biology-10-00652-f007:**
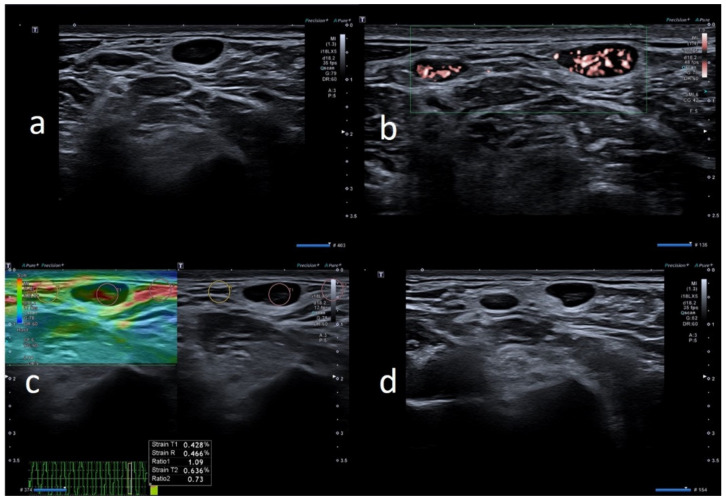
41-year-old male with palpable unilateral sopraclavicolar adenopathy noted 3 days after receiving the first dose of the Moderna COVID-19 vaccine in his left deltoid muscle. (**a**) B-mode sonogram image shows 2 ovular hypoechoic lymph nodes with hilum absence. (**b**) SMI image shows central and peripheral vascularization (**c**) Elastonography strain shows similar pattern of the node compared to surrounding tissue. (**d**) After 1 month, the cortical thickening appears uniform with hilum evidence.

**Figure 8 biology-10-00652-f008:**
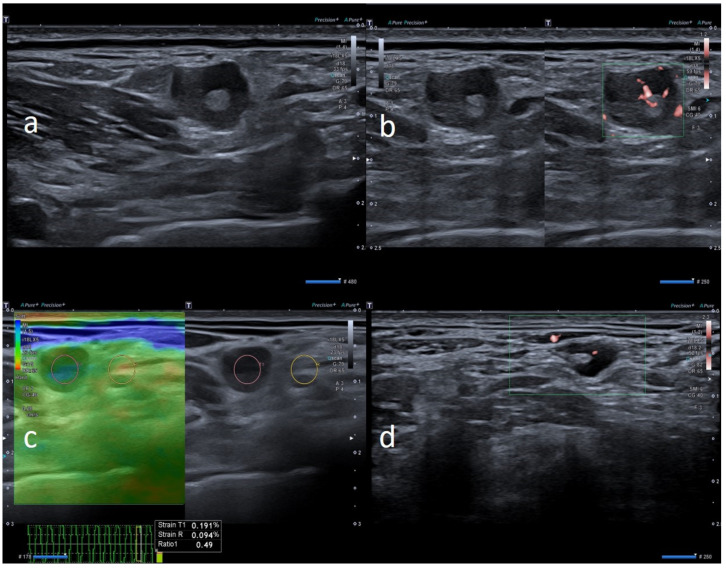
69-year-old female with palpable unilateral sopraclavicolar adenopathy noted 3 days after receiving the first dose of the AstraZeneca COVID-19 vaccine in her left deltoid muscle. (**a**) B-mode sonogram image shows round lymph node with asymmetric cortical thickening and hilum evidence. (**b**) SMI image shows central and peripheral vascularization. (**c**) Elastonography strain shows similar pattern of the node compared to surrounding tissue. (**d**) After 1 month, the cortical thickening appears uniform, with hilum evidence.

**Table 1 biology-10-00652-t001:** Patient study group: clinical characteristics and US features.

							US Features					
Sex	Age	Vaccine	Lymph Node Localization	Clinical Presentation	Oncological History	Nm	Size	Form	Cortical Thickening and Hilum	Sonoelasto	SMI	US Follow-Up to 2 Weeks
W	25	Pfizer	Axillary ipsilateral to vaccine injection	Three days after first dose of vaccine, axillary swelling and pain present. Also hypomobility ipsilateral arm	None	6	Variable: from 0.7 cm to 2.8 cm	Ovular	Prev. no hilum evidence	Prevalent hard pattern	Central and peripheral vascular signals	Normalized to 60 days
M	64	Pfizer	Supraclavicular ipsilateral to vaccine injection	Occasional autopalpation 2 weeks after second dose vaccine	None	2	Around 1.0 cm	Ovular	Assimetric cortical thickening with hilum evidence	Stiffness similar to surrounding tissue	Normal	Unnecessary other follow-up
W	28	Astazeneca	Supraclavicular ipsilateral to vaccine injection	Two days after first dose vaccine, supraclavicular swelling and pain present	None	3	Variable: from 0.6 to 1.5 cm	Ovular	No evidence hilum	Prevalent hard pattern	Central and peripheral vascular signals	Normalized to 45 days
W	72	Astazeneca	Supraclavicular ipsilateral to vaccine injection	One day after first dose vaccine, supraclavicular swelling and pain present	None	2	Subcentimetric size	Round	Asimmetric cortical thickening with hilum evidence	Prevalent hard pattern	Peripheral vascular signals	Normalized to 30 days
M	42	Pfizer	Axillary ipsilateral to vaccine injection	Occasionally 1 weeks after second dose during chest ct	None	3	Variable: from 1.5 to 2.0 cm	Ovular	Simmetric cortical thickening with normal hilum	Stiffness similar to surrounding tissue	Normal	Unnecessary other follow-up
M	39	Astazeneca	Supraclavicular ipsilateral to vaccine injection	Four days after first dose vaccine, supraclavicular swelling and pain present	None	1	Subcentimetric size	Round	No evidence hilum	Prevalent hard pattern	Central and peripheral vascular signals	Normalized to 30 days
M	60	Pfizer	Supraclavicular ipsilateral to vaccine injection	Occasional autopalpation 12 days after second dose vaccine	None	2	Around 1.0 cm	Ovular	Simmetric cortical thickening with normal hilum	Stiffness similar to surrounding tissue	Central and peripheral vascular signals	Unnecessary other follow-up
W	49	Pfizer	Axillary ipsilateral to vaccine injection	Occasionally, 6 days after first dose, during breast sonography for oncological surveillance	Breast cancer 3 years ago	4	Variable: from 1.0 to 2.0 cm	Ovular	Asimmetric cortical thickening and poor evidence hilum	Stiffness similar to surrounding tissue	Central and peripheral vascular signals	Unnecessary other follow-up
M	41	Moderna	Supraclavicular ipsilateral to vaccine injection	Two days after first dose vaccine, supraclavicular swelling and pain present	None	3	Subcentimetric size	Ovular	No evidence hilum	Stiffness similar to surrounding tissue	Central and peripheral vascular signals	Normalized to 30 days
W	54	Pfizer	Axillary ipsilateral to vaccine injection	Occasionally, 14 days after second dose, during breast sonography for surveillance	None	3	Variable: from 1.0 to 2.0 cm	Ovular	Simmetric cortical thickening with normal hilum	Stiffness similar to surrounding tissue	Normal	Unnecessary other follow-up
W	74	Astazeneca	Supraclavicular ipsilateral to vaccine injection	Ten days after first dose vaccine	None	2	Subcentimetric size	Ovular	Simmetric cortical thickening with normal hilum	Stiffness similar to surrounding tissue	Normal	Unnecessary other follow-up
M	35	Pfizer	Supraclavicular ipsilateral to vaccine injection	Day after the first dose vaccine, supraclavicular swelling and pain present	None	1	Around 1.5 cm	Round	No evidence hilum	Prevalent hard pattern	Central and peripheral vascular signals	Normalized to 60 days
W	52	Pfizer	Axillary ipsilateral to vaccine injection	Occasionally during breast sonography for oncological surveillance	None	3	Variable: from 1.0 to 2.5 cm.	Ovular	Simmetric cortical thickening with normal hilum	Stiffness similar to surrounding tissue	Normal	Unnecessary other follow-up
W	26	Astazeneca	Supraclavicular ipsilateral to vaccine injection	2 days after the first dose vaccine, axillary swelling and pain present	None	5	Subcentimetric size	Ovular	No hilum evidence	Prevalent hard pattern	Central and peripheral vascular signals	Normalized to 45 days
W	53	Pfizer	Axillary ipsilateral to vaccine injection	Occasionally, 16 days after first dose, during breast sonography for oncological surveillance	Breast cancer 2 years ago	3	Variable: from 1.0 to 2.0 cm	Ovular	Asimmetric cortical thickening with hilum evidence	Stiffness similar to surrounding tissue	Normal	Unnecessary other follow-up
M	62	Pfizer	Axillary ipsilateral to vaccine injection	Occasionally, 2 weeks after first dose, during chest ct to monitor small polmonary nodules	None	3	Variable: from 1.5 to 2.0 cm.	Ovular	Simmetric cortical thickening with normal hilum	Stiffness similar to surrounding tissue	Normal	Unnecessary other follow-up
M	57	Pfizer	Axillary ipsilateral to vaccine injection	Occasional autopalpation 2 weeks after second dose vaccine	Kidney cancer 4 years ago	2	Around 1.0 cm	Ovular	Asimmetric cortical thickening with hilum evidence	Stiffness similar to surrounding tissue	Normal	Normalized to 30 days
W	69	Astazeneca	Supraclavicular ipsilateral to vaccine injection	Three days after first dose vaccine	None	3	Subcentimetric size	Round	Asimmetric cortical thickening with hilum evidence	Stiffness similar to surrounding tissue	Central and peripheral vascular signals	Unnecessary other follow-up
W	37	Pfizer	Axillary ipsilateral to vaccine injection	Three days after first dose vaccine, axillary swelling present	Melanoma 5 years ago	5	Variable: from 1.5 to 2.0 cm.	Ovular	Assimetric cortical thickening with hilum evidence	Prevalent hard pattern	Central and peripheral vascular signals	Normalized to 45 days
M	63	Moderna	Axillary ipsilateral to vaccine injection	Occasionally, 16 days after first dose, during mammography	None	3	Variable: from 1.5 to 2.0 cm.	Ovular	Simmetric cortical thickening with normal hilum	Stiffness similar to surrounding tissue	Normal	Unnecessary other follow-up
F	32	Astrazeneca	Supraclavicular ipsilateral to vaccine injection	Day after first dose vaccine, supraclavicular swelling and pain present	None	1	Around 1.2 cm	Round	No evidence hilum	Prevalent hard pattern	Central and peripheral vascular signals	Normalized to 30 days
F	29	Pfizer	Supraclavicular ipsilateral to vaccine injection	Day after first dose vaccine, supraclavicular swelling and pain present	None	2	Subcentimetric	Round	No evidence hilum	Prevalent hard pattern	Central and peripheral vascular signals	Normalized to 30 days
F	66	Moderna	Supraclavicular ipsilateral to vaccine injection	Occasionally, 2 weeks after second dose, during shoulder rm	None	2	Around 1.0 cm	Ovular	Assimetric cortical thickening with hilum evidence	Stiffness similar to surrounding tissue	Central and peripheral vascular signals	Normalized to 30 days
M	59	Astrazeneca	Supraclavicular ipsilateral to vaccine injection	Occasionally, autopalpation 4 days after second dose vaccine	None	2	0.7 and 1.2 cm	Ovular	Asimetric cortical thickening with hilum evidence	Stiffness similar to surrounding tissue	Normal	Unnecessary other follow-up

## Data Availability

The datasets generated during and/or analyzed during the current study are not publicly available due to the clinical and confidential nature of the material but can be made available from the corresponding author on reasonable request.

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
