# Peer review of "Lymphadenopathy after the Anti-COVID-19 Vaccine: Multiparametric Ultrasound Findings"

_biology, 2021, doi:10.3390/biology10070652_

Round 1

Reviewer 1 Report

This is a retrospective study describing the ultrasound characteristics of 66 lymph nodes from a series of 24 patients, who developed symptomatic or radiologically-abnormal lymphadenopathy after COVID vaccination. The main result was that the lymphadenopathy demonstrated "worrisome" ultrasound findings.

Some queries:
- How were the ultrasound measurements done, standardized and validated?
- It is unclear why the findings are worrisome, especially given the clear post-vaccination context. How do the ultrasound characteristic compare with lymphadenopathy from patients who did not receive COVID vaccination but had other pathology?
- It is unclear if pathology has been definitively ruled out. Was FNA or biopsy done, especially when the lymph nodes were deemed to be abnormal? How was skin/soft tissue infection secondary to injection ruled out?
- It is unclear from the presentation of results how the ultrasound findings are similar among the 3 types of COVID vaccines used 
- It is unclear how the conclusion that ultrasound findings help to recognize the clinical scenario came about. Could it be that the ultrasound findings actually lead to confusion rather than clarity?

Author Response

Thank you for considering our article and to revise this manuscript accordingly. Kind regards.

Reviewer 2 Report

The manuscript entitled " Lymphadenopathy after the anti-COVID-19 vaccine: multiparametric ultrasound findings". Title, abstract and overall rationale of work to some extent is very nice and novel. However, there are still some minor concerns, which needs to be addressed and needs minor revision.

  1. The content of the manuscript is too short, specifically introduction and material methods section. Moreover, mechanism section is not properly elaborated, which is important for the reproducibility of the research.

  2. I would suggest the authors to enhance your theoretical discussion and arrives your debate or argument.
  3. A flowchart should be added to this research article to show the clear methodology and mechanism.

  1. Conclusion section must be elaborated.

Author Response

Thank you for considering our article and to revise this manuscript accordingly. Kind regards

Reviewer 3 Report

This is an interesting experience in a common-clinical practice problem during covid vaccination.

The awareness of this condition could be cost-saving for the health-care system and reduce patients anxiety.

Some considerations:

- If these data are available, it would be interesting to understand if these patients presented with anamnestic characteristics that could be associated with adenopathies development. Did you have in your series oncologic/oncohematologic patients or patients with autoimmune diseases?

- As no particular morphologic US characteristics emerged (some patients showed reactive US pattern, other "worrying" features) it would be useful in the Discussion or Conclusions to suggest a possible algorithm explaining which patients should be further followed with US basing on clinical and radiologic finding (e.g. patients with adenopathies <1.5 cm appeared shortly after covid vaccine and radiologically reactive, in my opinion should not perform further radiologic exams in the absence of clinical recurrence. Of course this could be not true in patients with oncologic/hematologic/autoimmune comorbidities)

- A table summarizing US patients characteristics would improve the manuscript quality making it more readible

Author Response

(The authors gave the same response as above.)

Reviewer 4 Report

Major comments

  1. Page 2 Section 2.2 last sentence. Can the authors explain the reason for dividing the elastosonography pattern into soft and hard and what criterion was used to differ hard from soft?
  2. Page 11 second paragraph. “Oncology patients are generally advised to be vaccinated against COVID-19, particularly because they are at higher risk of dying from COVID-19 than the general population.” Are there any references to support this statement?
  3. Maybe the authors want to discuss more about their research findings in the Discussion instead of majorly citing research work from others.

Minor comments:

  1. Page 2 Section 2.1 ling 5. Should “post-antiCOVID-19” be “post-anti-COVID-19”? Similar issue in page 10 Section 4 line 7.
  2. Page 2 Section 2.1 last sentence. Please indicate which Figure is referred and a period is missing.
  3. Page 2 Section 2.2. Are “symmetric” and “asymmetric” typos?
  4. Page 3 last sentence. “Figure” should be “Figures”.
  5. Figures are not shown properly in current layout. Please adjust it.
  6. Periods are missing from the caption of figures.

Author Response

(The authors gave the same response as above.)
